# PATCorrect: Non-autoregressive Phoneme-augmented Transformer for ASR Error Correction

## Abstract

Speech-to-text errors made by automatic speech recognition (ASR) system negatively impact downstream models relying on ASR transcriptions. Language error correction models as a post-processing text editing approach have been recently developed for refining the source sentences. However, efficient models for correcting errors in ASR transcriptions that meet the low latency requirements of industrial grade production systems have not been well studied. In this work, we propose a novel non-autoregressive (NAR) error correction approach to improve the transcription quality by reducing word error rate (WER) and achieve robust performance across different upstream ASR systems. Our approach augments the text encoding of the Transformer model with a phoneme encoder that embeds pronunciation information. The representations from phoneme encoder and text encoder are combined via multi-modal fusion before feeding into the length tagging predictor for predicting target sequence lengths. The joint encoders also provide inputs to the attention mechanism in the NAR decoder. We experiment on 3 open-source ASR systems with varying speech-to-text transcription quality and their erroneous transcriptions on 2 public English corpus datasets. Results show that our PATCorrect (Phoneme Augmented Transformer for ASR error Correction) consistently outperforms state-of-the-art NAR error correction method on English corpus across different upstream ASR systems. For example, PATCorrect achieves 11.62% WER reduction (WERR) averaged on 3 ASR systems compared to 9.46 % WERR achieved by other method using text only modality and also achieves an inference latency comparable to other NAR models at tens of millisecond scale, especially on GPU hardware, while still being 4.2 - 6.7x times faster than autoregressive models on Common Voice and LibriSpeech datasets.

## 1 Introduction

Automatic speech recognition (ASR) models transcribe human speech into readable text. It has many applications including real-time captions and meeting transcriptions. ASR model is also a critical component in large-scale natural language processing (NLP) systems like Amazon Alexa, Google Home and Apple Siri. Transcribed text serves as input for downstream models such as intent detection in voice assistants and response generation in voice chatbots. Errors made in speech-to-text ASR transcriptions can severely impact the accuracy of downstream models and thus lower the performance of the entire NLP system.

Recent advances in ASR systems using Transformer Gulati et al. (2020); Tüske et al. (2021) and CNN based models Li et al. (2019) have achieved state-of-the-art (SOTA) accuracy as measured by word error rate (WER). However, due to the complexity of human natural language and the quality of speech audios, even SOTA ASR systems can still make unavoidable and unrecoverable errors such as phonetic confusion between similar-sounding expressions. To improve the quality of ASR transcriptions, error correction models are applied to the outputs from ASR systems to detect and correct errors.

ASR error correction can be formulated as a sequence-to-sequence generation task, taking the ASR transcribed text as input source sequence and the ground-truth speech-to-text transcription as target

sequence. Previous studies D'Haro & Banchs (2016); Liao et al. (2020); Mani et al. (2020) have proposed sequence-to-sequence models that decode the target sequence in an autoregressive (AR) manner. Wang et al. (2020) added phoneme information to the AR decoder and found that it helps retrieve the correct entity from ASR transcriptions. These autoregressive models achieve SOTA accuracy but incur high latency making them infeasible for online production systems with low-latency constraints. For example, for voice digital assistants the end-to-end latency for a response is at the order of milliseconds for high quality user experience. Hence when incorporating such error correction models into the whole system, we need to seriously consider the speed and accuracy trade-off. Autoregressive decoding is a big bottleneck as it cannot be parallelized during inference, which does not meet the latency buffer allocated to the ASR error correction component in the end-to-end pipeline. Therefore, the critical need of reducing latency brings us the strong motivation to use non-autoregressive (NAR) models over AR models. Leng et al. (2021) applied a NAR sequence generation model with edit alignment to Chinese corpus that achieved comparable WER reduction and is 6 times faster than AR models. However, the performance of this NAR approach has not been tested for English corpus.

In this paper, we propose PATCorrect (Phoneme Augmented Transformer for ASR error Correction) as shown in Figure 1, a novel NAR based ASR error correction model with edit alignment that is based on both text and phoneme representations of the ASR transcribed sentences. PATCorrect creates inputs for the length tagging predictor by applying a multi-modal fusion approach to combine phoneme representation and text representation into joint feature embeddings. Both encoders (text and phoneme) interact with NAR decoder via encoder-decoder attention mechanism. PATCorrect improves the WER reduction (WERR) to 11.62% compared to the FastCorrect which is the SOTA NAR method that solely uses text only representation of the input, with comparable inference latency at tens of milliseconds scale. PATCorrect model is robust and scalable to different upstream ASR systems. We use three ASR systems to transcribe two public English corpus datasets, LibriSpeech and Common Voice, respectively to get their erroneous transcriptions as inputs. Experimental evaluations demonstrate that applying PATCorrect can consistently improve the transcription WER across different upstream ASR models with varying levels of transcription quality. To demonstrate our performance improvement, we benchmark against other ASR error correction models by applying them to the same sets of erroneous transcriptions.

Our contributions are summarized as follows:

• We propose PATCorrect, a novel model based on the Transformer architecture for NAR ASR correction. This model uses a multi-modal fusion approach that augments the traditional input text encoding with an additional phoneme encoder to incorporate pronunciation information, which is one of the key characteristics for spoken utterances.

• Through extensive offline evaluations, We demonstrate that PATCorrect outperforms the state-of-the-art NAR ASR error correction model that uses text only modality. For example, PATCorrect improves WERR to 11.62% with an inference latency at the same tens of milliseconds scale, while still being about 4.2 - 6.7x times faster than AR models.

• To the best of our knowledge, we are the first to establish that multi-modal fusion is a promising direction for improving the accuracy of low latency NAR methods for ASR error correction, and comprehensively study the performance of NAR ASR error correction for English corpus across different ASR systems with varying levels of quality.

## 2 RELATED WORK

### AUTOREGRESSIVE METHODS

The goal of ASR error correction is to convert erroneous source sequences from ASR outputs to target sequences with errors corrected. It can be viewed as Neural Machine Translation (NMT) problem with erroneous sentences as source language, and corrected sentences as target language. Therefore, research on ASR error correction started with conventional statistical machine translation methods. Cucu et al. (2013) applied it in domain-specific ASR systems for error correction. Anantaram et al. (2018) further utilized ontology learning to repair ASR outputs by a 4-step method. Recent NMT methods based on Transformers Vaswani et al. (2017); Ng et al. (2019) have become

increasingly accurate and have inspired applications to ASR error correction Liao et al. (2020); Mani et al. (2020); Hu et al. (2020). Based on the intuition that phonetic information helps with understanding ASR errors Fang et al. (2020); Sundararaman et al. (2021), Wang et al. (2020) found that adding phoneme information for domain-agnostic ASR system could benefit entity retrieval task. Although achieving high accuracy, these encoder-decoder based autoregressive generation models suffer from slow inference speed, error propagation and demand for a large amount of training data.

NON-AUTOREGRESSIVE METHODS

To address these issues in AR models, non-autoregressive sequence generation methods in NMT Gu et al. (2017), which aim to speed up the inference of AR models while maintaining comparable accuracy, has been a popular research topic in recent years. For NAR decoder, the length predictor is crucial as it outputs latent variables to determine the target sequence length for parallel generation. Gu et al. (2019) proposed to use dynamic insertion/deletion to iteratively refine the generated sequences based on previous predictions. Ghazvininejad et al. (2019) used a conditional masked language modeling for more efficient iterative parallel decoding. Straight-forward adaptation of these NAR methods from machine translation to the ASR error correction problem may even lead to increase in WER as shown in Leng et al. (2021). Current SOTA method for Chinese corpus error correction Leng et al. (2021) utilized edit alignment with text editing operations (*i.e.* insertion, deletion, substitution) to train the length predictor and assign each source token with a length tag.

## 3 PATCORRECT FOR ASR ERROR CORRECTION

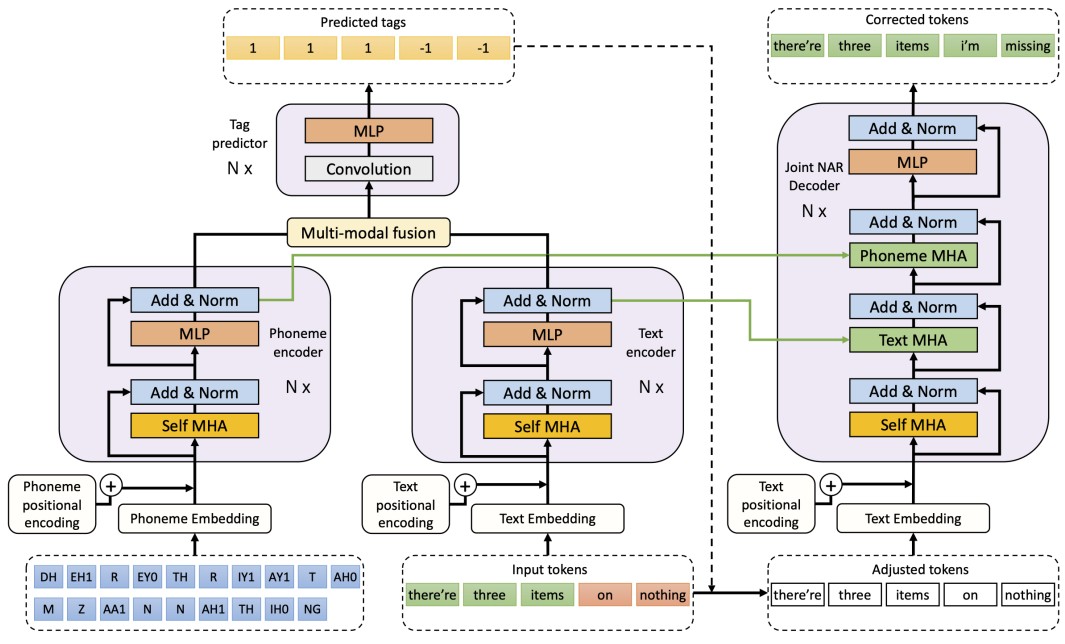

Figure 1: The overview of PATCorrect architecture. Starting from the bottom, we input the text sequence $\boldsymbol{w}$ and phoneme sequence $\boldsymbol{p}$ to two encoders separately. The outputs from phoneme and text encoders are combined together by multi-modal fusion operations, and then fed into the tag predictor $TagP$ for adjusting source tokens. We use two encoder-decoder attention layers sequentially in the joint NAR decoder for parallel decoding, to get the target sequences with ASR error correction.

The goal of ASR error correction is to correct the erroneous source tokenized sentence $\boldsymbol{w} = \{w_1, w_2, ..., w_n\}$ to the target error-free sequence $\hat{\boldsymbol{w}} = \{\hat{w}_1, \hat{w}_2, ..., \hat{w}_{\hat{n}}\}$, where the length of input tokens $n$ and the length of output tokens $\hat{n}$ can be the same or different. In PATCorrect, we add the phoneme sequence $\boldsymbol{p} = \{p_1, p_2, ..., p_m\}$ of the source sentence $\boldsymbol{w}$ to represent the pronunciation information. During training, we first pre-compute the text editing path from source sequence $\boldsymbol{w}$ to

target sequence $\hat{w}$ in the training set, similar to Leng et al. (2021). Then both source sequence $w$ and corresponding phoneme sequence $p$ are used as inputs to train a tag predictor which generates token-level alignment tags $t = \{t_1, t_2, ..., t_n\}$. For each source token $w_i$, the corresponding alignment tags $t_i$ consist of 4 possible edit operation types: $t_i = 1$ means keep the token unchanged; $t_i = 0$ means delete this token; $t_i = -1$ means substitute the token with another token; $t_i < -1$ means add other tokens adjacent to this token, for example, $t_i = -5$ means add 4 adjacent tokens. Before input into the non-autoregressive (parallel) decoder, the source tokens are adjusted based on the corresponding alignment tags in order to match the target sequence. During inference when we do not have ground truth target sequence to compute text editing paths, the tag predictor is used to predict the edit tags for the input source sequence, and then the target sequence is generated according to predicted tags.

## 3.1 Phoneme augmented encoder

The vanilla Transformer model Vaswani et al. (2017) consists of a single encoder and a single decoder module, each of which is a stack of several identical layers. We augment the architecture by adding an additional encoder to provide more information, which conceptually could encode any information modality depending on the application. Here we choose to use phoneme sequence because homophone error is one of the major sources of ASR speech-to-text transcription errors Fang et al. (2020); Sundararaman et al. (2021). The two encoders first encode text and phoneme information separately without sharing model parameters. The stacked encoder layers, consisting of multi-head self-attention layers and position-wise fully connected feed-forward (MLPs) layers, transform the input text sequence $w$ and phoneme sequence $p$ into hidden representations $H_w = \{h_{w_1}, h_{w_2}, ...h_{w_n}\} \in \mathbb{R}^{n \times d_h}$ and $H_p = \{h_{p_1}, h_{p_2}, ...h_{p_m}\} \in \mathbb{R}^{m \times d_h}$, with the output dimension $d_h$. We use the same hidden dimension $d_h$ for both encoders to simplify the multi-modal fusion operations later. There are two purposes for $H_w$ and $H_p$: firstly their fused representations are input into tag predictor for predicting the edit alignment corresponding to each source token, secondly they both provide inputs to encoder-decoder cross attention mechanism in the joint NAR decoder during parallel decoding.

## 3.2 Multi-modal fusion for tag predictor

The tag predictor $TagP$ is trained using precomputed ground truth tags with optimal editing alignment paths from source sequences to target sequences Leng et al. (2021). The text encoder output $H_w$ and phoneme encoder output $H_p$ are combined into $H_s$ via multi-modal fusion operations before feeding into the tag predictor to get one-dimensional scalar vector $t \in \mathbb{R}^{n \times 1}$, that has the same length $n$ as the source sequence. We experiment with 3 different fusion approaches Kiela et al. (2018); Ghazvininejad et al. (2018) to combine the text encoder output and phoneme encoder output, and then compare their performances in Sec. 5.

**Concatenation** We concatenate the two encoder outputs $H_w$ and $H_p$ together so that $H_s = \{h_{w_1}, h_{w_2}, ...h_{w_n}, h_{p_1}, h_{p_2}, ...h_{p_m}\} \in \mathbb{R}^{(n+m) \times d_h}$. By feeding this fused representation to convolutional and linear layers, the tag predictor $TagP$ will output one intermediate vector $I \in (n+m) \times 1$ that has the same length as the fused input $H_s$. We then crop $I$ by selecting the first $n$ dimensions to get the final tag prediction $t \in \mathbb{R}^{n \times 1}$.

**Pooling operations** Considering that normally $n \neq m$ for these multi-modal inputs, we first zero-pad the smaller dimension encoder outputs to make them the same length. In this ASR error correction application where the length of phoneme sequences is longer than or equal to the length of the text sequences, $i.e.$ $m \geq n$, we pad $H_w \in \mathbb{R}^{n \times d_h}$ with zeros to create $H'_w \in \mathbb{R}^{m \times d_h}$. Then component-wise addition or max pooling can be applied to $H'_w$ and $H_p$:

$$H_s = \text{Addition}(H'_w, H_p) \tag{1}$$

or

$$H_s = \text{Max}(H'_w, H_p) \tag{2}$$

where $H_s \in \mathbb{R}^{m \times d_h}$. Similarly, we crop the vector $I \in \mathbb{R}^{m \times 1}$ which is the intermediate output of $TagP(H_s)$ by selecting the first $n$ dimensions to get the final tag prediction $t \in \mathbb{R}^{n \times 1}$.

**Cross attention** We add cross attention layers with learnable parameters to project the phoneme encoder outputs on the text encoder outputs, which can also handle the embedding length difference

between the different modalities. We compute the cross attention outputs by taking text encoder output $H_{\boldsymbol{w}}$ as query, phoneme encoder output $H_{\boldsymbol{p}}$ as key and value:

$$H_{\boldsymbol{s}} = \text{Softmax}(\frac{(H_{\boldsymbol{w}}W_c^Q)(H_{\boldsymbol{p}}W_c^K)^T}{\sqrt{d_h}})(H_{\boldsymbol{p}}W_c^V) \tag{3}$$

where $W_c^Q, W_c^K, W_c^V \in \mathbb{R}^{d_h \times d_h}$ are the parameter matrices in cross attention layer with multiple attention heads, $H_{\boldsymbol{s}} \in \mathbb{R}^{n \times d_h}$ is the output after additional dropout, residual connection and normalization layers. No further cropping operation is needed since $TagP(H_{\boldsymbol{s}}) = \boldsymbol{t} \in \mathbb{R}^{n \times 1}$.

### 3.3 Non-autoregressive joint decoder

In the conventional AR model, the next token in the output sequence is conditioned on previous predicted tokens to form a chain of conditional probabilities:

$$P_{\text{AR}}(\hat{\boldsymbol{w}}|\boldsymbol{w};\theta) = \prod_{i=1}^{\hat{n}+1} P(\hat{w}_i|\hat{w}_{0:i-1}, w_{1:n};\theta) \tag{4}$$

These conditional probabilities can be trained autoregressively with cross-entropy loss at each decoding step. Adapted from Gu et al. (2017), our NAR model utilizes the tag predictor output as a latent variable to indicate the target length beforehand. So the conditional probability of the target sequence $\hat{\boldsymbol{w}}$ is defined as:

$$P_{\text{PATCorrect}}(\hat{\boldsymbol{w}}|\boldsymbol{w}, \boldsymbol{p};\theta) = \sum_{\boldsymbol{t} \in \mathcal{T}} \left( \prod_{i=1}^{n} P_{TagP}(t_i|\boldsymbol{w}, \boldsymbol{p};\theta) \cdot \prod_{i=1}^{\hat{n}} P(\hat{w}_i|\boldsymbol{w}_{\boldsymbol{t}}', \boldsymbol{p};\theta) \right) \tag{5}$$

where $\mathcal{T}$ is the set of all possible tag sequences that produces the correct editing of target sequences of length $\hat{n}$, and $\boldsymbol{w}_{\boldsymbol{t}}'$ is the adjusted source inputs based on predicted tags $\boldsymbol{t}$. The adjusted source sequence $\boldsymbol{w}_{\boldsymbol{t}}'$ is the actual input to the NAR decoder. Therefore, the training of PATCorrect is equivalent to optimizing the overall maximum likelihood with a variational lower bound:

$$\mathcal{L}_{\text{PATCorrect}} = \log P_{\text{PATCorrect}}(\hat{\boldsymbol{w}}|\boldsymbol{w}, \boldsymbol{p};\theta)$$

$$= \log \left( \mathop{\mathbb{E}}_{\boldsymbol{t} \sim q} \left( \prod_{i=1}^{n} P_{TagP}(t_i|\boldsymbol{w}, \boldsymbol{p};\theta) \cdot \prod_{i=1}^{\hat{n}} P(\hat{w}_i|\boldsymbol{w}_{\boldsymbol{t}}', \boldsymbol{p};\theta) \frac{1}{q(\boldsymbol{t})} \right) \right)$$

$$\geq \mathop{\mathbb{E}}_{\boldsymbol{t} \sim q} \left( \sum_{i=1}^{n} \log P_{TagP}(t_i|\boldsymbol{w}, \boldsymbol{p};\theta) + \sum_{i=1}^{\hat{n}} \log P(\hat{w}_i|\boldsymbol{w}_{\boldsymbol{t}}', \boldsymbol{p};\theta) \right) + \mathbb{H}(q) \tag{6}$$

where $\mathbb{H}(q) = -\mathbb{E}_{\boldsymbol{t} \sim q}(\log q(\boldsymbol{t}))$ is the Shannon entropy. With the optimal edit alignment paths, we provide an approximate distribution $q$ for the tag sequence with the most likely sample as the expectation. The first term within $\mathbb{E}_{\boldsymbol{t} \sim q}(\cdot)$ trains the tag predictor $TagP$, and the second term trains the error correction model. This design enables the parallel decoding for every target token $\hat{w}_i$ at the same time.

In the meantime, we include the encoder-decoder attention mechanism for both of our encoders in the joint NAR decoder, by sequentially combining them together Wang et al. (2020). After getting the self-attention output from decoder $H_{\hat{\boldsymbol{w}}} \in \mathbb{R}^{\hat{n} \times d_h}$ with causal mask removed to enable parallel calculation similar to Gu et al. (2017), we calculate the text and phoneme encoder-decoder attention by first using $H_{\hat{\boldsymbol{w}}}$ as query, text encoder output $H_{\boldsymbol{w}}$ as key and value:

$$Z_w = \text{Softmax}(\frac{(H_{\hat{\boldsymbol{w}}}W_w^Q)(H_{\boldsymbol{w}}W_w^K)^T}{\sqrt{d_h}})(H_{\boldsymbol{w}}W_w^V) \tag{7}$$

and then using the output $Z_w$ as query, phoneme encoder output $H_{\boldsymbol{p}}$ as key and value:

$$Z_p = \text{Softmax}(\frac{(Z_w W_p^Q)(H_{\boldsymbol{p}}W_p^K)^T}{\sqrt{d_h}})(H_{\boldsymbol{p}}W_p^V) \tag{8}$$

where $\{W_w^Q, W_w^K, W_w^V\}, \{W_p^Q, W_p^K, W_p^V\} \in \mathbb{R}^{d_h \times d_h}$ are the parameter matrices in the text and phoneme encoder-decoder multi-head attention layers, respectively. The remaining of the decoder layers are the same as the vanilla Transformer model where $Z_p \in \mathbb{R}^{\hat{n} \times d_h}$ will be input into fully-connected MLPs with residual connection and layer normalization.

## 4 EXPERIMENTS

### 4.1 DATASETS AND ASR MODELS

We create training data specifically designed for correcting ASR errors because ASR systems are accurate for most tokens and source sentences are usually aligned monotonically with target sentences unlike the shuffle error in NMT. For the ASR transcription dataset $\mathbb{S}^{\text{trans}}$, multiple ASR systems $\{\mathcal{A}_k\}_{k \leq K}$ are used to transcribe public English audio corpus $\mathbb{A} = \{\mathcal{C}, \mathcal{S}_{gt}^{\text{trans}}\}$, where the audio clips $\mathcal{C}$ have corresponding ground truths from human transcriptions $\mathcal{S}_{gt}^{\text{trans}}$. The transcriptions of $\mathcal{C}$ from ASR systems $\mathcal{S}_f^{\text{trans},k} = \mathcal{A}_k(\mathcal{C})$ contain actual ASR errors, which leads to the ASR transcription dataset $\mathbb{S}^{\text{trans}} = \bigcup_{k=1}^K \{\mathcal{S}_f^{\text{trans},k}, \mathcal{S}_{gt}^{\text{trans}}\} = \bigcup_{k=1}^K \{\mathcal{A}_k(\mathcal{C}), \mathcal{S}_{gt}^{\text{trans}}\}$. We combine two public English datasets, LibriSpeech Panayotov et al. (2015) and Common Voice v9.0 Ardila et al. (2019), together to get corpus $\mathbb{A}$.

In order to test different ASR systems with different architecture and performances, we choose 3 pre-trained ASR systems $\{\mathcal{A}_k\}_{k \leq 3}$ implemented in NeMo Kuchaiev et al. (2019): The Convolution-augmented Transformer **Conformer** Gulati et al. (2020)[1] trained on several thousands hours of English speech with top-tier performance; The Convolution-based **Jasper** Li et al. (2019)[2] trained on 7,057 hours of audio samples with above average performance; A light-weight 5x5 **QuartzNet** Kriman et al. (2020)[3] trained on 960 hours LibriSpeech dataset with subpar performance. All 3 ASR systems are trained with the Connectionist Temporal Classification (CTC) loss. We use the default splits in LibriSpeech and Common Voice with transcriptions from 3 ASR systems together to compose the dataset $\mathbb{S}^{\text{trans}}$ as shown in Table 1, which in total has more than 3.5 million sentences pairs in training split. Note that LibriSpeech has been included in the training of all 3 ASR systems, so we use DEV and TEST splits from Common Voice as benchmarks in accuracy evaluations later.

Table 1: ASR transcription dataset statistics and ASR original WER

| | LibriSpeech | | | Common Voice | | |
|---|---|---|---|---|---|---|
| | TRAIN | DEV | TEST | TRAIN | DEV | TEST |
| # of sents | 281,241 | 5,567 | 5,559 | 890,107 | 16,331 | 16,318 |
| Avg. words/sent | 33.4 | 18.9 | 18.9 | 10.3 | 9.8 | 9.4 |
| Conformer WER | 1.58 | 3.21 | 3.27 | 7.62 | 9.84 | 10.03 |
| Jasper WER | 2.13 | 6.87 | 6.90 | 13.63 | 18.91 | 21.02 |
| QuartzNet WER | 1.97 | 10.37 | 11.01 | 36.42 | 47.90 | 52.85 |

### 4.2 EVALUATION METRICS

The performance of ASR error correction models should be evaluated in two aspects: accuracy and speed. Speed is measured by the latency of the whole inference process including encoding and decoding for different methods. For accuracy, we use Word Error Rate (WER), WER Reduction (WERR), $F_{0.5}$, $F_{0.25}$, and $Correction$. Similar to Leng et al. (2021), WER is defined as total edit distances[4] between source and target divided by the total number of words in target sequence, and WERR is defined as the percentage of improvement in WER. For error detection ability, $Precision$ measures how many actual error tokens are edited among all of the edited tokens, *i.e.*, how many of them actually need to be corrected; $Recall$ measures how many actual error tokens are edited among all of the error tokens. In this application of ASR error correction, unnecessary edits may even lead to WER increase because original ASR outputs are mostly correct. So to put more weight on $Precision$, we use $F_{0.5}$ and $F_{0.25}$ as overall measurements for error detection ability as in Liao et al. (2020); Omelianchuk et al. (2020); Rothe et al. (2021). For error correction ability, $Correction$,

---

[1] https://catalog.ngc.nvidia.com/orgs/nvidia/teams/nemo/models/stt_en_conformer_ctc_large

[2] https://catalog.ngc.nvidia.com/orgs/nvidia/teams/nemo/models/stt_en_jasper10x5dr

[3] 'QuartzNet5x5LS-En': https://catalog.ngc.nvidia.com/orgs/nvidia/models/nemospeechmodels

[4] total edit distances = total number of substitutions + insertions + deletions

defined as $(\text{\# of correctly edited error tokens})/(\text{\# of edited error tokens})$, measures the percentage of edited error tokens that match the ground truth.

## 4.3 MODEL CONFIGURATIONS

**PATCorrect**   In our PATCorrect architecture, we use 6-layer text encoder, 6-layer phoneme encoder and 6-layer joint decoder with the hidden model dimension $d_h = 512$, and MLP dimension $d_{\text{MLP}} = 2048$. We use 8 attention heads for self-attention, encoder-decoder attention and cross attention respectively. In the cross attention setup for fusing two encoder outputs, we use 2 consecutive modules of cross attention with dropout, residual connection and layer normalization. For tag predictor $TagP$, we apply 5 layers of convolutional modules which consists of 1-D Convolution layer with kernel size equals 3, ReLU activation, layer normalization and dropout, followed by 2 layers of MLPs to generate one-dimensional vector $t$ which is the output from tag predictor $TagP$. We use MSE loss to train $TagP$ to predict the length adjustment for each source token.

**Baseline models**   We compare our model with both AR Transformer and popular NAR methods. For **AR Transformer**, we use the standard vanilla Transformer architecture with 6-layer encoder and 6-layer decoder and the same hidden size $d_h = 512$. For NAR methods from NMT, we use the default training hyperparameters from Mask Predict (**CMLM**) Ghazvininejad et al. (2019) and Levenshtein Transformer (**LevT**) Gu et al. (2019)[5]. For SOTA NAR method for ASR error correction problem **FastCorrect** Leng et al. (2021)[6], we adapt the same 6-layer encoder-decoder architecture and the same architecture for its length predictor.

**Training and inference details**   All models are implemented using Fairseq Ott et al. (2019), and trained using 4 NVIDIA Tesla V100 GPUs with maximum batch token size of 5000 and label smoothed cross entropy loss function. They are trained from scratch using the ASR transcription dataset $\mathbb{S}^{\text{trans}}$ for 30 epochs. Source and target sentences are tokenized using sentencepiece Kudo & Richardson (2018), and the phoneme sequences are generated by English grapheme to phoneme conversion using the CMU pronouncing dictionary[7], an independent post-processing step of the 1-best ASR hypothesis as model inputs. We use Adam optimizer Kingma & Ba (2014) and inverse square root for learning rate scheduling starting from $5e^{-4}$. During inference, we set the test batch size as 1 to simulate the automated machine learning system environment with output from the upstream ASR system, and all NAR methods have the same max decoding iteration of 1. Different hardware conditions are tested including single NVIDIA Tesla V100 GPU, 8 CPUs and 4 CPUs with Intel(R) Xeon(R) CPU E5-2686 v4 @ 2.30GHz.

## 5 RESULTS

### 5.1 ACCURACY

We compare the WER and WERR using different error correction models for all three ASR systems and their total combined transcriptions on Common Voice DEV and TEST splits, as shown in Table 2 and 3. The lower WER and higher WERR, the better model accuracy. Results show: 1) In total, the proposed PATCorrect model beats the SOTA FastCorrect method by improving the TEST set WERR from 9.46 to 11.62, which is more than 20% relative improvement; 2) Our PATCorrect model outperforms other NAR methods robustly when dealing with all three ASR system transcriptions. Straight-forward adaptations of NMT NAR methods like CMLM and LevT may even introduce more WER for some well-performing ASR systems like Conformer and Jasper; 3) Among all of the multi-modal fusion operations experimented, cross attention performs best across almost all datasets.

### 5.2 SPEED

We test inference speed on both GPU and CPUs as shown in Table 4. The inference latency on LibriSpeech (LS) is longer than that of Common Voice (CV) because LibriSpeech has twice average

---

[5]https://github.com/facebookresearch/fairseq/tree/main/examples/nonautoregressive_translation
[6]https://github.com/microsoft/NeuralSpeech/tree/master/FastCorrect
[7]https://github.com/Kyubyong/g2p

Table 2: WER (%) ↓ using different error correction models

| Models | Conformer | | Jasper | | QuartzNet | | Total | |
|---|---|---|---|---|---|---|---|---|
| | DEV | TEST | DEV | TEST | DEV | TEST | DEV | TEST |
| No error correction | 9.84 | 10.03 | 18.91 | 21.02 | 47.90 | 52.85 | 25.55 | 27.96 |
| AR Transformer | 8.56 | 9.26 | 15.95 | 18.56 | 33.84 | 40.64 | 19.45 | 22.82 |
| CMLM | 10.43 | 10.72 | 19.25 | 21.37 | 42.09 | 47.39 | 23.92 | 26.49 |
| LevT | 10.05 | 10.37 | 18.69 | 20.74 | 40.82 | 45.98 | 23.19 | 25.70 |
| FastCorrect | 9.25 | 9.88 | 17.45 | 19.96 | 39.69 | 46.11 | 22.13 | 25.32 |
| PATCorrect($cat$) | 9.22 | **9.78** | 17.45 | 19.93 | 39.61 | 46.00 | 22.09 | 25.24 |
| PATCorrect($add$) | 9.26 | 9.83 | 17.50 | 19.96 | 39.67 | 46.04 | 22.14 | 25.28 |
| PATCorrect($max$) | 9.20 | 9.79 | 17.34 | 19.82 | 39.25 | 45.60 | 21.93 | 25.07 |
| PATCorrect($cross\_atten$) | **9.15** | 9.84 | **17.28** | **19.80** | **38.27** | **44.50** | **21.57** | **24.72** |

Table 3: WERR (%) ↑ using different error correction models

| Models | Conformer | | Jasper | | QuartzNet | | Total | |
|---|---|---|---|---|---|---|---|---|
| | DEV | TEST | DEV | TEST | DEV | TEST | DEV | TEST |
| AR Transformer | 13.07 | 7.65 | 15.64 | 11.70 | 29.34 | 23.10 | 23.87 | 18.40 |
| CMLM | -6.00 | -6.89 | -1.80 | -1.67 | 12.12 | 10.32 | 6.36 | 5.26 |
| LevT | -2.15 | -3.41 | 1.15 | 1.35 | 14.77 | 12.99 | 9.24 | 8.11 |
| FastCorrect | 5.98 | 1.41 | 7.68 | 5.05 | 17.13 | 12.74 | 13.37 | 9.46 |
| PATCorrect($cat$) | 6.29 | **2.44** | 7.73 | 5.17 | 17.31 | 12.95 | 13.53 | 9.75 |
| PATCorrect($add$) | 5.90 | 1.94 | 7.45 | 5.04 | 17.18 | 12.88 | 13.33 | 9.61 |
| PATCorrect($max$) | 6.50 | 2.32 | 8.28 | 5.71 | 18.05 | 13.71 | 14.16 | 10.35 |
| PATCorrect($cross\_atten$) | **7.01** | 1.81 | **8.61** | **5.80** | **20.11** | **15.79** | **15.59** | **11.62** |

words per sentence as shown in Table 1. Consistent with the observation from Gu et al. (2017), AR model shows a linear latency trend with decoding lengths, while the latency of the NAR methods only slightly increases. PATCorrect achieves an inference latency which is comparable with other NAR models, especially on GPU hardware it is only slower than FastCorrect, while still being about 4.2 - 6.7x times faster than AR models on Common Voice and LibriSpeech datasets.

Table 4: Inference latency ↓ using different hardwares with unit ms/sentence

| Models | Single GPU | | 8×CPUs | | 4×CPUs | |
|---|---|---|---|---|---|---|
| | LS | CV | LS | CV | LS | CV |
| AR Transformer | 246.74 | 141.00 | 306.40 | 149.79 | 366.54 | 186.89 |
| CMLM | 39.49 | 35.54 | 47.96 | 43.78 | 55.15 | 47.90 |
| LevT | 52.61 | 47.86 | 49.89 | 45.68 | 60.27 | 55.33 |
| FastCorrect | **22.25** | **20.81** | **29.28** | **23.77** | **36.02** | **29.14** |
| PATCorrect($cross\_atten$) | 36.92 | 33.71 | 52.47 | 41.89 | 66.97 | 49.97 |

## 5.3 SENSITIVITY ANALYSIS

To further investigate the performance improvements of PATCorrect, we conduct sensitivity analysis by comparing the error detection ability and error correction ability for different models with 3 ASR system transcriptions in total. In Table 5, $P, R, C$ denote $Precision, Recall, Correction$ respectively. Results show that our PATCorrect not only has the highest $F_{0.5}$ and $F_{0.25}$ score with great $Precision$ that is comparable to AR model, but also has better ability to edit error tokens to the correct targets indicated by higher $Correction$. As shown in the examples in Appendix Table 7, our PATCorrect's design of adding a phoneme information helps our model identify and correct the sound-alike ASR errors that previous method couldn't correct with only text information.

Table 5: Sensitivity metrics ↑ for error correction models

| Models | DEV | | | | | TEST | | | | |
|---|---|---|---|---|---|---|---|---|---|---|
| | $P$ | $R$ | $F_{0.5}$ | $F_{0.25}$ | $C$ | $P$ | $R$ | $F_{0.5}$ | $F_{0.25}$ | $C$ |
| AR Transformer | 91.50 | 66.72 | 85.17 | 89.54 | 44.53 | 90.89 | 65.33 | 84.30 | 88.85 | 37.81 |
| CMLM | 83.39 | 61.70 | 77.92 | 81.70 | 31.43 | 84.87 | 61.45 | 78.86 | 83.01 | 27.37 |
| LevT | 85.08 | **61.95** | 79.17 | 83.26 | 32.53 | 86.49 | **61.75** | 80.07 | 84.49 | 28.42 |
| FastCorrect | 90.25 | 61.64 | 82.58 | 87.85 | 32.66 | 89.53 | 60.90 | 81.83 | 87.12 | 27.70 |
| PATCorrect ($cross\_atten$) | **91.42** | 60.28 | **82.86** | **88.72** | **34.45** | **90.27** | 59.64 | **81.86** | **87.62** | **29.50** |

## 5.4 ABLATION STUDY

We perform ablation study to understand the effectiveness of different components of PATCorrect by removing a component while retaining the others. No phoneme input for $TagP$ means that we only use text information as input for predicting token tags while still using phoneme encoder-decoder attention in NAR decoder. No phoneme attention means that we remove phoneme encoder-decoder attention from the NAR decoder and only use text encoder-decoder attention, while still using cross-attention to fuse the text and phoneme encoder outputs for $TagP$. We also increase the amount of pre-training data by using a synthetic dataset $\mathbb{S}^{\text{synth}}$ for data augmentation to pretrain the model for 20 epochs. We crawl 50M sentences from English Wiki pages[8], and add random editing operations like deletion, insertion, substitution with a homophone dictionary to produce erroneous sentences paired with the original correct texts, that mimics the ASR errors with a simulated error distribution. The results in Table 6 with WER and WERR, equal-weight averaged from all 3 ASR systems, show that adding phoneme information in $TagP$ and NAR decoder lead to better WER and WERR. Our PATCorrect, using phoneme information in both tag predictor and NAR decoder, yields the best results. In addition, using synthetic dataset for pretraining can also boost the model accuracy to further reduce WER.

Table 6: Ablation study for PATCorrect model

| Models | DEV | | TEST | |
|---|---|---|---|---|
| | WER ↓ | WERR ↑ | WER ↓ | WERR ↑ |
| No error correction | 25.55 | - | 27.96 | - |
| FastCorrect | 22.13 | 13.37 | 25.32 | 9.46 |
| No phoneme input for $TagP$ | 21.88 | 14.35 | 25.13 | 10.14 |
| No phoneme attention in decoder | 21.90 | 14.30 | 25.06 | 10.39 |
| PATCorrect($cross\_atten$) | 21.57 | 15.59 | 24.72 | 11.62 |
| PATCorrect($cross\_atten$) + $\mathbb{S}^{\text{synth}}$ | **21.12** | **17.35** | **24.33** | **13.00** |

## 6 CONCLUSIONS

We propose PATCorrect, a novel NAR phoneme-augmented Transformer-based model with robust performance on different upstream ASR systems with varying speech-to-text transcription quality, to serve as an independent post-processing text editing component for reducing the errors in ASR transcriptions. Our model outperforms state-of-the-art NAR ASR error correction models, and still 4.2 - 6.7x times faster than AR models, which makes it a great fit as industrial scale text editing method to refine ASR transcriptions. Our study establishes that multi-modal fusion is a promising direction for improving the accuracy of low latency NAR methods for ASR error correction. Future explorations including adding context information for source sequences, and using knowledge distillation with AR model as teacher Zhou et al. (2019) to improve the performance of PATCorrect.

---

[8]https://github.com/attardi/wikiextractor

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

## A   APPENDIX

We show some examples in Common Voice TEST split with ground truths, QuartzNet transcriptions, FastCorrect and PATCorrect corrections.

Table 7: Examples for error correction sentences

| Ground truths | ASR transcriptions | FastCorrect corrections | PATCorrect corrections |
|---|---|---|---|
| i don't blame you | i don't lame you | i don't lame you | i don't blame you |
| areabased visualizations have existed for decades | areobased visualizations have existed for decates | a realbased visualizations have existed for decays | areabased visualizations have existed for decades |
| a blond woman is singing on stage | a blond woman is singing on states | a blond woman is singing on sticks | a blond woman is singing on stage |
| a white bike is leaning against a post | a white wike is leaning against the post | a white white is leaning against the post | a white bike is leaning against a post |
| he attended carnegie mellon university | he attended canegy melon niversity | he attended carnegie milan university | he attended carnegie mellon university |
| make sure you get a doctor's note | make sure you get a doctor's nowte | make sure you get a doctor's know it | make sure you get a doctor's note |
| he was the prime suspect out of thirteen | he was the prime suspect out of thirte n | he was the prime suspect out of the team | he was the prime suspect out of thirteen |
| we need to run | wanet to run | what it to run | we need to run |
| she was silent for a moment or two | she was silent for a moment or to | she was silent for a moment or too | she was silent for a moment or two |
| it was one of the first private commercial broadcaster in the netherlands | it was one of the furest private commercial brod gaster in the nidderlans | it was one of the first private commercial broadcaster in the midlands | it was one of the first private commercial broadcaster in the netherlands |
| they are available in shades of white pink purple and blue | they are available in shades of white pak purple and blue | they are available in shades of white pig purple and blue | they are available in shades of white pink purple and blue |
| the bloom of the rose lasts a few days | the bloom of the rors lasts a few days | the bloom of the rows lasts a few days | the bloom of the rose lasts a few days |
| a man in a blue shirt is sitting at a bus stop | a man in a blue shore tos sitting at a busttup | a man in a blue shirt is sitting at a bus | a man in a blue shirt is sitting at a bus stop |

