# OpenReview forum: "PATCorrect: Non-autoregressive Phoneme-augmented Transformer for ASR Error Correction"
_ICLR.cc/2023/Conference — Submitted to ICLR 2023_

### Official Review · Reviewer_XMid · 2022-10-21

**Confidence:** 4
**Correctness:** 3
**Technical Novelty And Significance:** 3
**Empirical Novelty And Significance:** 3
**Recommendation:** 6

**Clarity, Quality, Novelty And Reproducibility:**

* Mostly quite clear.  One clarification question: Where does the phone sequence used for correction come from?  Is it generated from the ASR system directly, or is it a post-processing of the ASR hypothesis?  Also, Is there any benefit from consuming an N-best list of hypotheses from the ASR models?  While not all ASR models generate phone sequences, some do (or can be trained to simultaneously with words).  All decoders can produce N-best lists.

Since the models explored are full-context models (i.e. not streaming), it's not clear that latency is critical to their performance.  Low latency error correction would be more compelling on a streaming model.

Quality: Strong
Novelty: Sufficient Novelty
Reproducibility: Could reproduce.

**Strength And Weaknesses:**

Strength
* The use of phonetic information in error correction is well motivated.
* The architecture is well described and well motivated.

Weakness
* The improvements offered by PATCorrect are somewhat modest.  There is a small, but consistent improvement over FastCorrect.  But the performance is still substantially worse than AR Transformer.
* The reporting of performance is, at some points, strained.  1) The decision to use F_0.25 instead of a more common detection metric of F_1 is surprising, but the authors gesture toward the bias toward Precision.  However, this decision is less convincing when observing that FactCorrect has a better F_1 score than PATCorrect, though PATCorrect has a higher F_0.25 score.   2) The reported improvement of 20% in the abstract and introduction is a relative improvement to the relative improvement to WER (the primary metric for ASR).  This is not clear in the abstract.  The absolute WER difference between FastCorrect and PATCorrect is between 1 and 2%. WERR (relative improvement to WER) may be a more relevant measure for error correction, however, since this is already a relative measure, absolute difference to WERR are a more interpretable description of performance differences.  The underlying work is strong, but these presentation decisions leave the impression that the work is presented in an overly optimistic manner.

**Summary Of The Paper:**

PATCorrect is a non-autoregressive (NAR) error correction module.  PATCorrect is distinguished by using aligned phoneme and word inputs for making correction decisions.  Evaluation across a number of input ASR models demonstrates that PATCorrect outperforms FastCorrect, a similar NAR correction model which does not use phone information.

**Summary Of The Review:**

The central contribution of the work is strong, and clearly described.  The evaluations themselves are good.  However, the presentation could be more tempered to more accurately demonstrate the strengths and weaknesses of the approach.  A stronger motivation for the use of NAR error correction will also provide additional context for the contribution of the work.

---

> ### Author Response · Authors · 2022-11-15
> **Responses to Reviews of Reviewer XMid (Part 1/2)**
>
> We thank the reviewer for acknowledging our contributions and glad to agree on that our paper is well motivated with comprehensive evaluations. For your remaining concerns, we addressed them in detail as follows:
>
> > **C1:** The improvements offered by PATCorrect are somewhat modest. There is a small, but consistent improvement over FastCorrect. But the performance is still substantially worse than AR Transformer.
>
> **Answer:** The fundamental design of NAR [1] proposes the conditional independence assumption in the output sequence. Therefore, NAR is more difficult than AR which explicitly models the dependency in the target space. Our goal is to close the accuracy gap between NAR and AR methods, due to superior latency performance of NAR methods. PATCorrect is the best performing NAR model on tens of milliseconds scale of latency. AR method has high accuracy, but its latency is on hundreds of milliseconds scale. This latency difference is crucial in industry.
>
> [1] Gu, Jiatao, James Bradbury, Caiming Xiong, Victor OK Li, and Richard Socher. "Non-autoregressive neural machine translation." arXiv preprint arXiv:1711.02281 (2017).
>
> > **C2:** 1) The decision to use F_0.25 instead of a more common detection metric of F_1 is surprising, but the authors gesture toward the bias toward Precision. However, this decision is less convincing when observing that FactCorrect has a better F_1 score than PATCorrect, though PATCorrect has a higher F_0.25 score.
>
> **Answer:** Thank you for pointing this out. We chose F_0.25 to put more weight on the Precision metric, which is more important for user facing applications because we do not want to introduce more errors in the end-to-end task. Similar rationale for preferring Precision for text error correction as shown in [2-4] is by using F_0.5 score (as known as MaxMatch score). We use F_0.25 compared to F_0.5, because we want to be extremely conservative regarding false positives. In the revised manuscript, we have also added the F_0.5 comparison in Table 5.
>
> [2]. Liao, Junwei, Sefik Emre Eskimez, Liyang Lu, Yu Shi, Ming Gong, Linjun Shou, Hong Qu, and Michael Zeng. "Improving readability for automatic speech recognition transcription." arXiv preprint arXiv:2004.04438 (2020).
>
> [3]. Omelianchuk, Kostiantyn, Vitaliy Atrasevych, Artem Chernodub, and Oleksandr Skurzhanskyi. "GECToR--grammatical error correction: tag, not rewrite." arXiv preprint arXiv:2005.12592 (2020).
>
> [4]. Rothe, Sascha, Jonathan Mallinson, Eric Malmi, Sebastian Krause, and Aliaksei Severyn. "A simple recipe for multilingual grammatical error correction." arXiv preprint arXiv:2106.03830 (2021).
>
> > **C3:** 2) The reported improvement of 20% in the abstract and introduction is a relative improvement to the relative improvement to WER (the primary metric for ASR). This is not clear in the abstract. The absolute WER difference between FastCorrect and PATCorrect is between 1 and 2%. WERR (relative improvement to WER) may be a more relevant measure for error correction, however, since this is already a relative measure, absolute difference to WERR are a more interpretable description of performance differences. The underlying work is strong, but these presentation decisions leave the impression that the work is presented in an overly optimistic manner.
>
> **Answer:** Thank you for the thoughtful comment on using the appropriate presentation. In the revised abstract and introduction, we present the absolute difference in WERR, that PATCorrect achieves 11.62% WERR averaged on 3 ASR systems compared to 9.46 % WERR achieved by FastCorrect.
>
> (Due to the length limit, this is part 1 out of total 2 responses.)

---

> ### Author Response · Authors · 2022-11-15
> **Responses to Reviews of Reviewer XMid (Part 2/2)**
>
> > **C4:** One clarification question: Where does the phone sequence used for correction come from? Is it generated from the ASR system directly, or is it a post-processing of the ASR hypothesis? Also, Is there any benefit from consuming an N-best list of hypotheses from the ASR models? While not all ASR models generate phone sequences, some do (or can be trained to simultaneously with words). All decoders can produce N-best lists.
>
> **Answer:** In Sec. 4.3 “Training and inference details” of the revised manuscript, we clarified that the phoneme sequences are generated by English grapheme to phoneme conversion using the CMU pronouncing dictionary [5]. Therefore, it is a post-processing of the ASR transcription.
>
> Thank you for the inspiring idea of using an N-best list of hypotheses from the ASR models. Currently in our problem setup, we only have access to the 1-best hypothesis of these upstream black-box ASR systems used in this work. Since our phoneme sequences are generated from an independent post-processing step, our approach is generalizable and can be applied to any upstream ASR systems. Besides, we also tested the performance of PATCorrect on different upstream ASR systems with different tiers of performance. In the future work, we hope to utilize N-best candidates [6] or using k-NN memorization [7] to further close the accuracy gap between PATCorrect and AR models.
>
> [5]. https://github.com/Kyubyong/g2p
>
> [6]. Leng, Yichong, Xu Tan, Rui Wang, Linchen Zhu, Jin Xu, Wenjie Liu, Linquan Liu et al. "Fastcorrect 2: Fast error correction on multiple candidates for automatic speech recognition." arXiv preprint arXiv:2109.14420 (2021).
>
> [7]. Bekal, Dhanush, Ashish Shenoy, Monica Sunkara, Sravan Bodapati, and Katrin Kirchhoff. "Remember the context! ASR slot error correction through memorization." In 2021 IEEE Automatic Speech Recognition and Understanding Workshop (ASRU), pp. 236-243. IEEE, 2021.
>
> > **C5:** Since the models explored are full-context models (i.e. not streaming), it's not clear that latency is critical to their performance. Low latency error correction would be more compelling on a streaming model.
>
> **Answer:** Our target application field is, indeed, a streaming pipeline scenario. For example, for voice digital assistants the end-to-end latency for a response is at the order of milliseconds for high quality user experience. Hence when incorporating such error correction models into the whole system, we need to seriously consider the speed and accuracy trade-off. Autoregressive decoding is a big bottleneck as it cannot be parallelized during inference, which does not meet the latency buffer allocated to the ASR error correction component in the end-to-end pipeline. Therefore, the critical need of reducing latency brings us the strong motivation to use NAR models over AR models. In the 3rd paragraph of Introduction section of the revised manuscript, we add this discussion to provide additional context for our contribution.
>
> > **C6:** The central contribution of the work is strong, and clearly described. The evaluations themselves are good. However, the presentation could be more tempered to more accurately demonstrate the strengths and weaknesses of the approach. A stronger motivation for the use of NAR error correction will also provide additional context for the contribution of the work.
>
> **Answer:** Thank you again for acknowledging our contribution. The presentation concern is addressed in C2, C3 and the motivation of NAR is addressed in C5.
>
> (Due to the length limit, this is part 2 out of total 2 responses.)

---

### Official Review · Reviewer_9fDp · 2022-10-23

**Confidence:** 5
**Correctness:** 2
**Technical Novelty And Significance:** 2
**Empirical Novelty And Significance:** 2
**Recommendation:** 3

**Clarity, Quality, Novelty And Reproducibility:**

Where does the phoneme sequence come from? It seems just from the recognized word sequence. So no pronunciation variants? Also, why this way, why not use the ASR encoder output or so? Also, how relevant are the phonemes anyway? There is a bit of ablation study on this, but this comes very short, and was only done for some of the very weak models. It should be done for Conformer.

Related work: It only addresses error correction. But just shallow fusion / log-linear combination with a standard language model (LM), how is that really different? As far as I know, not much really can be gained from such error correction models over standard LM fusion with a good LM.

In general, this is really an important comparison, to make use of an external LM, in comparison to such error correction, or in combination, or other variations. Further, it makes sense to test different model architectures, such as CTC, Transducer or Attention-based Encoder-Decoder.

It seems the error correction model is just trained on transcriptions, not more. So this is much weaker than a LM trained on text-only data, which is usually available in much larger quantities. But the error correction model could maybe also trained on text-only data? This should be addressed.

It's even not clear, those pretrained models, are those Transducer, CTC, att-based enc-dec?
How is decoding done? Unclear.

Pretrained models, but not explained how pretrained or which pretrained models exactly? Are they public?

Speed comparison: I don't really know: Does this only measure PATCorrect itself, or the whole decoding? What is actually the underlying model and decoding procedure?

Latex wrong math usage, use \operatorname for all whole-word-functions like softmax, Addition, Max, etc.
Latex wrong math usage, use \textrm or so for text subscripts like PATCorrect etc.


**Details Of Ethics Concerns:**

Ok.

**Strength And Weaknesses:**

Strengths:

- An interesting novel model for error correction.
- I think the use of phoneme inputs is novel.

Weaknesses:

- No comparison to a standard language model fusion? See below.
- The code is not published?
- The analysis and ablations study is too short.
- Ablation studies only with weak model.
- Only tested on speech recognition.
- Many aspects are unclear. See below.
- Studies are on Common Voice, but LibriSpeech would be better, as this is what most people know much better.


**Summary Of The Paper:**

The task is automatic speech recognition (ASR). The paper proposed PATCorrect, a new correction model which operates on the output of another speech recognition model and tries to improve the output by correcting errors.

The proposed model works in a non-autoregressive way and thus can run in parallel, efficiently on hardware like GPUs.

It uses three different pretrained models to test the correction model on. The best model is a Conformer, which gets 10.03% WER on CommonVoice Test without correction, and the best PATCorrect correction gets down to 9.78% WER. The other models are very weak in comparison, but then the relative improvements are larger.

**Summary Of The Review:**

There are too many weaknesses, as explained above. This needs more work.

In general, this paper seems more appropriate for a speech conference (Interspeech etc)? While in principle it might be applicable to other tasks, I think this should be tested, and I think this is necessary for publication on ICLR. The novelty and scope is otherwise clearly not high enough. Otherwise, even for speech conferences, I think the quality is not high enough yet, and this needs to address the weaknesses I explained before.

---

> ### Author Response · Authors · 2022-11-15
> **Responses to Reviews of Reviewer 9fDp (Part 1/3)**
>
> We thank the reviewer for the constructive revision comments to which we furnish our responses on a point-by-point basis and revised our manuscript accordingly.
>
> > **C1:** No comparison to a standard language model fusion? See below.
> >
> > Related work: It only addresses error correction. But just shallow fusion / log-linear combination with a standard language model (LM), how is that really different? As far as I know, not much really can be gained from such error correction models over standard LM fusion with a good LM.
> >
> > In general, this is really an important comparison, to make use of an external LM, in comparison to such error correction, or in combination, or other variations. Further, it makes sense to test different model architectures, such as CTC, Transducer or Attention-based Encoder-Decoder.
> >
> > It seems the error correction model is just trained on transcriptions, not more. So this is much weaker than a LM trained on text-only data, which is usually available in much larger quantities. But the error correction model could maybe also trained on text-only data? This should be addressed.
>
> **Answer:** Using shallow fusion with an external LM on the decoder side is also an autoregressive architecture, which suffers from the high inference latency issue. In our related work comparison, we selected the most commonly used AR model: the attention-based encoder-decoder transformer. The focus of our work is to close the accuracy gap between NAR and AR models while retaining the latency strength in NAR models.
>
> Additionally, as for adapting the LM approach in the NAR manner, we have compared PATCorrect with CMLM [1], a NAR method using parallel decoding for masked language modeling. CMLM adapts a standard encoder-decoder transformer architecture, and the results show that PATCorrect performs better.
>
> Regarding the different model architectures, we used 3 ASR systems (Conformer [2], QuartzNet [3] and Jasper [4]) that have different model architectures and ASR error characteristics. Conformer uses Convolution-augmented Transformer architecture. QuartzNet’s design is based on the Jasper architecture, which is a convolutional model. The main novelty in QuartzNet’s architecture is that the 1D convolutions are replaced with 1D time-channel separable convolutions, an implementation of depth-wise separable convolutions. All 3 ASR systems are trained with the Connectionist Temporal Classification (CTC) loss.
>
> Regarding the training data, our PATCorrect is trained on text-only transcriptions instead of audio clips. In our ablation study Table 6, we crawled 50M sentences from English Wiki as text-only data and add random editing operations to synthesize training samples. The results show that using larger quantities of synthetic data could help improve the performance.
>
> [1]. Ghazvininejad, Marjan, Omer Levy, Yinhan Liu, and Luke Zettlemoyer. "Mask-predict: Parallel decoding of conditional masked language models." arXiv preprint arXiv:1904.09324 (2019).
>
> [2]. Gulati, Anmol, James Qin, Chung-Cheng Chiu, Niki Parmar, Yu Zhang, Jiahui Yu, Wei Han et al. "Conformer: Convolution-augmented transformer for speech recognition." arXiv preprint arXiv:2005.08100 (2020).
>
> [3]. Kriman, Samuel, Stanislav Beliaev, Boris Ginsburg, Jocelyn Huang, Oleksii Kuchaiev, Vitaly Lavrukhin, Ryan Leary, Jason Li, and Yang Zhang. "Quartznet: Deep automatic speech recognition with 1d time-channel separable convolutions." In ICASSP 2020-2020 IEEE International Conference on Acoustics, Speech and Signal Processing (ICASSP), pp. 6124-6128. IEEE, 2020.
>
> [4]. Li, Jason, Vitaly Lavrukhin, Boris Ginsburg, Ryan Leary, Oleksii Kuchaiev, Jonathan M. Cohen, Huyen Nguyen, and Ravi Teja Gadde. "Jasper: An end-to-end convolutional neural acoustic model." arXiv preprint arXiv:1904.03288 (2019).
>
> > **C2:** The code is not published?
>
> **Answer:** Publishing the PATCorrect code needs additional legal corporate approval, and we could not get in due to the time. We could make efforts to publish the code upon the acceptance of the paper.
>
> (Due to the length limit, this is part 1 out of total 3 responses.)

---

> ### Author Response · Authors · 2022-11-15
> **Responses to Reviews of Reviewer 9fDp (Part 2/3)**
>
> > **C3:** The analysis and ablations study is too short. Ablation studies only with weak model.
> >
> > Where does the phoneme sequence come from? It seems just from the recognized word sequence. So no pronunciation variants? Also, why this way, why not use the ASR encoder output or so? Also, how relevant are the phonemes anyway? There is a bit of ablation study on this, but this comes very short, and was only done for some of the very weak models. It should be done for Conformer.
>
> **Answer:** In Sec. 5.4 of the revised manuscript, we clarified the fact that we did ablation study on all 3 ASR systems, including Conformer. Table 6 shows the equally weighted average WER and WERR for the 3 ASR systems.
>
> In Sec. 4.3 “Training and inference details” of the revised manuscript, we clarified the phoneme sequences are generated by English grapheme to phoneme conversion using the CMU pronouncing dictionary [5]. We chose this design because in our current problem setup we can only get the 1-best hypothesis from the upstream black-box ASR system. We do not have the ASR encoder outputs available for our training.
>
> The usage of phoneme is very relevant and critical. FastCorrect, as a text-only NAR model, failed to detect and correct some errors related to pronunciation as shown in the Appendix Table 7, while PATCorrect which incorporated the phoneme information performs better.
>
> [5]. https://github.com/Kyubyong/g2p
>
> > **C4:** Studies are on Common Voice, but LibriSpeech would be better, as this is what most people know much better.
>
> **Answer:** Thank you for noticing this point. We chose Common Voice as our evaluation dataset because it is not included in the training of the 3 public ASR systems used in this study. These pretrained ASR models are mostly using LibriSpeech for training, thus performing with very high accuracy and not many errors. It does not really make sense to still correct those transcriptions.
>
> > **C5:** It's even not clear, those pretrained models, are those Transducer, CTC, att-based enc-dec? How is decoding done? Unclear.
> >
> > Pretrained models, but not explained how pretrained or which pretrained models exactly? Are they public?
>
> **Answer:** In Sec 4.1 of the revised manuscript, we further clarified the pretrained ASR models and added more details with references (Also described in responses to C1) [6-8]. In this work, we focus on the error correction model development, so we chose to use publicly available ASR pretrained models as upstream models for better reproducibility. It is worth noticing that our error correction models are all trained from scratch for fair comparison.
>
> [6]. https://catalog.ngc.nvidia.com/orgs/nvidia/teams/nemo/models/stt_en_conformer_ctc_large
>
> [7]. https://catalog.ngc.nvidia.com/orgs/nvidia/teams/nemo/models/stt_en_jasper10x5dr
>
> [8].’QuartzNet5x5LS-En’: https://catalog.ngc.nvidia.com/orgs/nvidia/models/nemospeechmodels
>
> > **C6:** Speed comparison: I don't really know: Does this only measure PATCorrect itself, or the whole decoding? What is actually the underlying model and decoding procedure?
>
> **Answer:** In Sec 4.2 of the revised manuscript, we clarified the procedure of speed comparison. To fairly compare every error correction model, the speed is measured for the whole inference process (encoding + decoding) for different methods. We used the publicly available official implementations for the comparing methods [9-10].
>
> The decoding procedure for PATCorrect is similarly described in [11-12]. We removed the causal mask used in the self-attention module of the conventional Transformer’s decoder and used the length information predicted from our TagP. In this way, we can pre-determine the target sequence length and decode every target token at the same time to greatly reduce the inference latency.
>
> [9]. https://github.com/facebookresearch/fairseq/tree/main/examples/nonautoregressive_translation
>
> [10]. https://github.com/microsoft/NeuralSpeech/tree/master/FastCorrect
>
> [11]. Gu, Jiatao, James Bradbury, Caiming Xiong, Victor OK Li, and Richard Socher. "Non-autoregressive neural machine translation." arXiv preprint arXiv:1711.02281 (2017).
>
> [12]. Leng, Yichong, Xu Tan, Linchen Zhu, Jin Xu, Renqian Luo, Linquan Liu, Tao Qin, Xiangyang Li, Edward Lin, and Tie-Yan Liu. "Fastcorrect: Fast error correction with edit alignment for automatic speech recognition." Advances in Neural Information Processing Systems 34 (2021): 21708-21719.
>
> (Due to the length limit, this is part 2 out of total 3 responses.)

---

> ### Author Response · Authors · 2022-11-15
> **Responses to Reviews of Reviewer 9fDp (Part 3/3)**
>
> > **C7:** Only tested on speech recognition.
> >
> > In general, this paper seems more appropriate for a speech conference (Interspeech etc)? While in principle it might be applicable to other tasks, I think this should be tested, and I think this is necessary for publication on ICLR. The novelty and scope is otherwise clearly not high enough. Otherwise, even for speech conferences, I think the quality is not high enough yet, and this needs to address the weaknesses I explained before.
>
> **Answer:** Thank you for acknowledging that the proposed NAR architecture has the potential to be applied to other applications, that involves information from multiple modalities, to bring latency improvement comparing with AR models. In this work, we focus on the ASR error correction task which is a rising application and provide valuable benchmarks for the English corpus across different ASR systems. We also formulate this work as a broad and generic text-editing problem, for example, PATCorrect uses text-only data for training regardless of upstream ASR models and transcription qualities, and PATCorrect is essentially a sequence-to-sequence approach with inputs from multiple modalities. Our study establishes that multi-modal fusion is a promising direction for improving the accuracy of low latency NAR methods for ASR error correction.
>
> We humbly believe this work should be interesting to ICLR audience working on NLP, ASR, or text-editing applications.
>
> > **C8:** Latex wrong math usage, use \operatorname for all whole-word-functions like softmax, Addition, Max, etc.
> Latex wrong math usage, use \textrm or so for text subscripts like PATCorrect etc.
>
> **Answer:** Thank you for pointing them out. In the revised manuscript, we corrected these wrong LaTex math usage as you suggested.
>
> (Due to the length limit, this is part 3 out of total 3 responses.)

---

### Official Review · Reviewer_RfZS · 2022-10-24

**Confidence:** 5
**Correctness:** 3
**Technical Novelty And Significance:** 2
**Empirical Novelty And Significance:** 2
**Recommendation:** 5

**Clarity, Quality, Novelty And Reproducibility:**

The paper is well written and the proposed multi-modal fusion method is easy to follow. But compared with FastCorrect, the novelty is quite limited.


**Strength And Weaknesses:**

Strength:

1.The idea of phoneme fusion and cross attention strategy is simple yet effective. The model outperforms FastCorrect on English datasets, and achieves SOTA results on English corpus.

2. The paper compares three feature fusion approaches, and shows that cross attention is effective for text and phoneme features.

3. Experimental results are based on comparisons on 3 English ASR systems to show the robustness of the proposed approaches.

Weakness：
1. A Phoneme MHA module is added behind the original Text MHA module in the Transformer decoder block, but the ablation study shows that without this phoneme attention, the model performance roughly remains the same as adding it. So with more calculation on the phoneme attention, the model barely gets any improvement from it.
2. The improvements on some datasets are minor. All of the fusion approaches of combining the phoneme embedding and text embedding don’t consider the relationship between each word and its corresponding pronunciation, which may be important for the words error correction.
3. The novelty of proposed framework is not significant. The architecture does not seem that there is too much difference with FastCorrect except the phoneme embedding, and two extra cross attention. Besides that, model architecture and train process are almost the same.
4. Phoneme feature is commonly used in error correction [1, 2], and from the ablation experiments we see a lot of contributions are from phoneme feature. To my knowledge cross-attention feature fusion can be seen in Image field previously [3].

[1] Wang et al., ASR Error Correction with Augmented Transformer for Entity Retrieval. Interspeech 2020.

[2] Fang et al., Non-Autoregressive Chinese ASR Error Correction with Phonological Training. NAACL 2022.

[3] Bai et al., TransFusion: Robust LiDAR-Camera Fusion for 3D Object Detection with Transformers. CVPR 2022.



**Summary Of The Paper:**

The paper proposes a new non-autoregressive model that incorporates the pronunciation information with text information for the task of ASR error correction. The proposed model uses a transformer architecture for two encoders and a decoder, it adopts three different fusion approaches to combine the representations from phoneme encoder and text encoder for the target length prediction. The decoder computes the cross attention with both encoders sequentially and predicts the target words.

**Summary Of The Review:**

This paper proposes a network of NAR Transformer with a phoneme encoder. Compared with FastCorrect, the improvement is limited, but it requires more latency than FastCorrect. Besides, the idea of incorporating phoneme information to gain better ability of error correction is useful but not very creative, since many text correction approaches have been proposed in recent years with similar ideas.

---

> ### Author Response · Authors · 2022-11-15
> **Responses to Reviews of Reviewer RfZS (Part 1/2)**
>
> We thank the reviewer for acknowledging our contributions and providing insightful suggestions. We revised our manuscript accordingly and the detailed responses are provided below to address your concerns:
>
> > **C1:** A Phoneme MHA module is added behind the original Text MHA module in the Transformer decoder block, but the ablation study shows that without this phoneme attention, the model performance roughly remains the same as adding it. So with more calculation on the phoneme attention, the model barely gets any improvement from it.
>
> **Answer:** In Sec. 5.4 of the revised manuscript, we re-organized the statements to avoid the confusion here. “No phoneme attention in decoder” means that we remove phoneme encoder-decoder attention from the NAR decoder and only use text encoder-decoder attention, while still using cross-attention to fuse the text and phoneme encoder outputs for predicting adjusted tokens (Tag Predictor input). “No phoneme attention in decoder” has a test WERR of 10.39 and “PATCorrect(cross_atten)” has WERR of 11.62. Comparing these two, we draw the conclusion that the model is improved with the phoneme attention calculation in the decoder.
>
> > **C2:** The improvements on some datasets are minor. All of the fusion approaches of combining the phoneme embedding and text embedding don’t consider the relationship between each word and its corresponding pronunciation, which may be important for the words error correction.
>
> **Answer:** As we experimented on 3 ASR systems with different capability of speech recognition, the error correction improvement is limited if the original ASR system already performs well without producing many errors. Moreover, PATCorrect consistently performs better than other NAR models, regardless of upstream ASR models.
>
> The cross-attention approach as the fusion method calculates the projection of the phoneme information on the text information, which can be considered intuitively as calculating the importance of each pronunciation on each word. Therefore, not only the relationship between each word and its corresponding pronunciation is included in the fused embeddings, but also the pronunciation of other words in the sentence. Compared with other fusion methods like Addition and MaxPooling [1] which also considered the relationship between each word and its phoneme, our results validate that the cross-attention fusion method works the best regarding the specific task of ASR error correction.
>
> [1]. Kiela, Douwe, Edouard Grave, Armand Joulin, and Tomas Mikolov. "Efficient large-scale multi-modal classification." In Proceedings of the AAAI Conference on Artificial Intelligence, vol. 32, no. 1. 2018.
>
> > **C3:** The novelty of proposed framework is not significant. The architecture does not seem that there is too much difference with FastCorrect except the phoneme embedding, and two extra cross attention. Besides that, model architecture and train process are almost the same.
>
> **Answer:** Our significant contributions compared with FastCorrect are as follows:
> 1. FastCorrect retains the original NAR architecture that was firstly proposed in [2]. We propose a new NAR architecture with an additional phoneme encoder considering the error characteristics in ASR transcriptions. We also experiment with several fusion methods for combining the embeddings from different modalities to give the best practices.
> 2. Without losing generality, the proposed NAR architecture has the potential to be applied to other applications, that involve information from multiple modalities, to bring latency improvement compared with AR models.
> 3. FastCorrect is only trained and tested on the Chinese corpus. Its performance remains unknown for the English corpus. We benchmark the results for multiple public ASR systems using the English corpus to draw the conclusion that PATCorrect performs consistently better than FastCorrect.
>
> [2]. Gu, Jiatao, James Bradbury, Caiming Xiong, Victor OK Li, and Richard Socher. "Non-autoregressive neural machine translation." arXiv preprint arXiv:1711.02281 (2017).
>
> (Due to the length limit, this is part 1 out of total 2 responses.)

---

> > ### Comment · Reviewer_RfZS · 2022-11-21
> > **Replies to Responses (part 1/2)**
> >
> > C1 Answer: In Sec. 5.4 of the revised manuscript, we re-organized the statements to avoid the confusion here. “No phoneme attention in decoder” means that we remove phoneme encoder-decoder attention from the NAR decoder and only use text encoder-decoder attention, while still using cross-attention to fuse the text and phoneme encoder outputs for predicting adjusted tokens (Tag Predictor input). “No phoneme attention in decoder” has a test WERR of 10.39 and “PATCorrect(cross_atten)” has WERR of 11.62. Comparing these two, we draw the conclusion that the model is improved with the phoneme attention calculation in the decoder.
> >
> > Reply: WERRs of 10.39 and 11.62 correspond to WERs 24.72 and 25.06. The gap seems not that much significant in WER.
> >
> > C2 Answer: As we experimented on 3 ASR systems with different capability of speech recognition, the error correction improvement is limited if the original ASR system already performs well without producing many errors. Moreover, PATCorrect consistently performs better than other NAR models, regardless of upstream ASR models.
> >
> > The cross-attention approach as the fusion method calculates the projection of the phoneme information on the text information, which can be considered intuitively as calculating the importance of each pronunciation on each word. Therefore, not only the relationship between each word and its corresponding pronunciation is included in the fused embeddings, but also the pronunciation of other words in the sentence. Compared with other fusion methods like Addition and MaxPooling [1] which also considered the relationship between each word and its phoneme, our results validate that the cross-attention fusion method works the best regarding the specific task of ASR error correction.
> >
> > [1]. Kiela, Douwe, Edouard Grave, Armand Joulin, and Tomas Mikolov. "Efficient large-scale multi-modal classification." In Proceedings of the AAAI Conference on Artificial Intelligence, vol. 32, no. 1. 2018.
> >
> > Reply: The difference with Adding and MaxPooling is compared, but my concern is the relationship between each word and its corresponding pronunciation which could improve the performance and is considered in Fang et al. NAACL 2022 [2].
> >
> > C3 Answer: Our significant contributions compared with FastCorrect are as follows:
> >
> > 1. FastCorrect retains the original NAR architecture that was firstly proposed in [2]. We propose a new NAR architecture with an additional phoneme encoder considering the error characteristics in ASR transcriptions. We also experiment with several fusion methods for combining the embeddings from different modalities to give the best practices.
> >
> > Reply: This didn't address my concern that "there is too much difference with FastCorrect except the phoneme embedding"
> >
> >
> > 2. Without losing generality, the proposed NAR architecture has the potential to be applied to other applications, that involve information from multiple modalities, to bring latency improvement compared with AR models.
> >
> > See reply to C5 Answer.
> >
> > 3. FastCorrect is only trained and tested on the Chinese corpus. Its performance remains unknown for the English corpus. We benchmark the results for multiple public ASR systems using the English corpus to draw the conclusion that PATCorrect performs consistently better than FastCorrect.
> >
> > See reply to C4 Answer.

---

> > > ### Author Response · Authors · 2022-12-01
> > > **2nd Round Replies to Responses**
> > >
> > > > **C1-R2:** WERRs of 10.39 and 11.62 correspond to WERs 24.72 and 25.06. The gap seems not that much significant in WER.
> > >
> > > **Answer:** In our proposed architecture, the phoneme information is not only used in the decoder block, but also in the encoder side and tag predictions. The gap mentioned above is only proving the effectiveness of using phoneme information in the decoder. The final improvement of our method, jointly from the tag predictions and additional phoneme attention calculation in the decoder, is significant comparing with FastCorrect. On the DEV set, we improve the WER from 22.13 to 21.57, with a WERR from 13.37 to 15.59. On the TEST set, we improve the WER from 25.32 to 24.72, with a WERR from 9.46 to 11.62. Because different datasets have different amounts of errors, WERR as a relative measurement is more comparable and reasonable to interpret.
> > >
> > > > **C2-R2:** The difference with Adding and MaxPooling is compared, but my concern is the relationship between each word and its corresponding pronunciation which could improve the performance and is considered in Fang et al. NAACL 2022 [2].
> > > >
> > > > [2] Fang et al., Non-Autoregressive Chinese ASR Error Correction with Phonological Training. NAACL 2022.
> > >
> > > **Answer:** Thank you for pointing us to this very recent reference, and it seems like an interesting phonological training approach. Comparing with the referred PhVEC method (Fang et al. NAACL 2022), we are not specially restricting the text with its own phoneme tokens, but using the cross attention mechanism to attend to different phoneme tokens from the entire sentence, which is a more general approach.
> > >
> > > Regarding the performance improvement in PhVEC, we also noticed that their evaluated ASR model for generating errors seems to be a single customized ASR model, which might not be comparable to publicly available ASR models. Their results might be limited to the error characteristics of the particular custom ASR model used in their study.
> > >
> > > (Responses to C3, C4 and C5 are addressed in another reply)

---

> ### Author Response · Authors · 2022-11-15
> **Responses to Reviews of Reviewer RfZS (Part 2/2)**
>
> > **C4:** Phoneme feature is commonly used in error correction [1, 2], and from the ablation experiments we see a lot of contributions are from phoneme feature. To my knowledge cross-attention feature fusion can be seen in Image field previously [3].
>
> **Answer:** We agree that utilizing phoneme feature indeed is essential for improving accuracy in ASR related applications as you mentioned. A key point is that we are the first to apply this intuition to the NAR architecture for ASR error correction and propose the cross-attention fusion approach to overcome the multi-modality input challenge. The experimental results also validate our novel design choices.
>
> > **C5:** The paper is well written and the proposed multi-modal fusion method is easy to follow. But compared with FastCorrect, the novelty is quite limited.
> >
> > This paper proposes a network of NAR Transformer with a phoneme encoder. Compared with FastCorrect, the improvement is limited, but it requires more latency than FastCorrect. Besides, the idea of incorporating phoneme information to gain better ability of error correction is useful but not very creative, since many text correction approaches have been proposed in recent years with similar ideas.
>
> **Answer:** The novelty concern comparing with FastCorrect is addressed in C3, the intuition of utilizing phoneme information and multi-modal fusion approach is addressed in C4. FastCorrect does not invent the NAR model architecture and the training process either, instead it adapts the idea into ASR error correction application. Our study establishes that multi-modal fusion is a promising direction for improving the accuracy of low latency NAR methods for ASR error correction.

---

> > ### Comment · Reviewer_RfZS · 2022-11-21
> > **Replies to Responses (part 2/2)**
> >
> > C4 Answer: We agree that utilizing phoneme feature indeed is essential for improving accuracy in ASR related applications as you mentioned. A key point is that we are the first to apply this intuition to the NAR architecture for ASR error correction and propose the cross-attention fusion approach to overcome the multi-modality input challenge. The experimental results also validate our novel design choices.
> >
> > Reply: As far as I know, Fang et al. NAACL 2022 [2] also use phoneme feature in NAR architecture, which shows better performance on the same Aishell test set tested in FastCorrect. For a fair and comprehensive comparison, I think PATCorrect should test the model on Aishell as well, because FastCorrect did some preprocess works for Chinese before training. Those works including large-scale pseudo data augmentation and edit distance alignment may be different in English from Chinese and have an influence on model performance.
> >
> > C5 Answer: The novelty concern comparing with FastCorrect is addressed in C3, the intuition of utilizing phoneme information and multi-modal fusion approach is addressed in C4. FastCorrect does not invent the NAR model architecture and the training process either, instead it adapts the idea into ASR error correction application. Our study establishes that multi-modal fusion is a promising direction for improving the accuracy of low latency NAR methods for ASR error correction.
> >
> > Reply: The answer didn't address my concern that the proposed model requires more latency than FastCorrect with limited improvement on WER.

---

> > > ### Author Response · Authors · 2022-12-01
> > > **2nd Round Replies to Responses**
> > >
> > > > **C4-R2:** As far as I know, Fang et al. NAACL 2022 [2] also use phoneme feature in NAR architecture, which shows better performance on the same Aishell test set tested in FastCorrect. For a fair and comprehensive comparison, I think PATCorrect should test the model on Aishell as well, because FastCorrect did some preprocess works for Chinese before training. Those works including large-scale pseudo data augmentation and edit distance alignment may be different in English from Chinese and have an influence on model performance.
> > >
> > > **Answer:**
> > > 1. As the PhVEC also mentioned in the last sentence of their paper *“As a future work, we plan to extend PhVEC to other languages and use corresponding phonological tokens to correct the variable length errors caused by pronunciation.”* This indicates that they only tested their method with one single corpus in Chinese, and their performances on different languages still remain unknown, which exactly suggests that our work in English corpus is valuable in the research community.
> > > 2. Our phoneme modality feature is more generic and available across different languages, comparing with the Pinyin phonological feature. With this paper as a foundation, we will evaluate our method in different languages in the future.
> > > 3. Our proposed cross-attention approach is a more generic way and applicable to different modalities, comparing with PhVEC that only sequentially insert the Pinyin tokens after the text tokens to combine them. As for the large-scale pseudo data augmentation and edit distance alignment, we also tested it in out paper and improved the results as shown in the last row of Table 6.
> > >
> > >
> > > > **C5-R2:** The answer didn't address my concern that the proposed model requires more latency than FastCorrect with limited improvement on WER.
> > >
> > > **Answer:** The reported latency results come from an experimental code where the dual encoders are computed sequentially, but we can optimize it by implementing them in parallel which is entirely possible by its design, to save latency in the final production system. This approach can easily scale to other different modalities. Additionally, the latency-accuracy trade-off shown in our paper opens new research directions for people to explore more possibilities to improve the current NAR approaches.

---

### Decision · Program_Chairs · 2023-01-20

**Decision:**

Reject

**Justification For Why Not Higher Score:**

There are lots of weakness in this paper as listed in the meta review. The authors didn't fully address the reviewers' concerns. The experiment baseline is not strong enough.

**Justification For Why Not Lower Score:**

N/A

**Metareview: Summary, Strengths And Weaknesses:**

This paper proposed to integrate phoneme information into the ASR error correction by fusing the phoneme information with the text information, under the non-autoregressive (NAR) framework.

Strength: The method is very well motivated, and the authors showed that the proposed method works well on 3 different types of pre-trained models, better than FastCorrect.

 Weakness:
The novelty is not significant. In the first round rebuttal, the authors argued they are the first to integrate phoneme information into the NAR error correction. However, after the reviewer pointed out a prior work also did so, then the authors switched their tone by claiming they applied the phoneme assisted error correction to English while the prior work was on Mandarin. To me, such application to a new language is not the novelty justifying the publication as ICLR.

While the authors answered most weakness challenges in their rebuttal, there are some answers not satisfying. For example, integrating acoustic information from encoder instead of using text/phoneme information should further improve correction accuracy. However, the authors didn't answer this challenge well.
Regarding the increased latency of the proposed method, the authors argued they are use experimental code, while in the future they should be able to optimize the latency. The answer is not satisfactory because we should judge the paper based on the current status instead of the promise of future.
The authors also stated in the rebuttal of not working on Librispeech is because the task has very low WER. This worries me a little because our hope is to further the best ASR model regardless the baseline WER. It seems the authors are picking some scenarios (e.g., training/testing mismatch) that have large room of improvement, and show the effectiveness of the proposed method on those scenarios.

The experiment baseline setup also has issues. The authors should have external LM fusion as a strong baseline to compare. But the authors just simply argued that external LM fusion is an AR method. However, the readers have more interests in the best error correction performance instead of whether the method is AR or NAR.
Furthermore, the authors used FastCorrect as the baseline. Why not the improved FastCorrect 2?
Leng, Yichong, et al. "Fastcorrect 2: Fast error correction on multiple candidates for automatic speech recognition." arXiv preprint arXiv:2109.14420 (2021).